# Mitigating the Negative Effect of Air Traffic Controller Mental Workload on Job Performance: The Role of Mindfulness and Social Work Support

**Bader Alaydi [1,*] and Siew-Imm Ng [2]**

1   Aviation & Management, College of Business Administration, Prince Sultan University, Riyadh 11586, Saudi Arabia
2   School of Business & Economics, Universiti Putra Malaysia, Serdang 43400, Malaysia
*   Correspondence: balaydi@psu.edu.sa

**Abstract:** Air traffic controllers (ATCOs) play a substantial part in securing the safety of flights, such that a compromise of the ATCOs' performance may lead to tragedies. Given the mental workload that comes with the nature of ATCOs' work, this study intends to investigate the impact of mental workload on ATCOs' job performance and identify conditional factors that could mitigate the mental workload–ATCOs' job performance relationship. Underpinned by the job demands–resources theory, a framework was developed to investigate the impact of job demands (mental workload) on job performance and whether personal resources (mindfulness and social work support) weaken the relationship. A total of 324 ATCOs across Saudi Arabia responded to the questionnaire. There were three notable findings. First, mental workload was indeed found to bring detrimental effects to ATCOs' job performance. Second, mindfulness played a mitigating role, where more mindful ATCOs demonstrated less workload effect on performance. Thirdly, social work support also played a mitigating role, where ATCOs who perceived receiving greater job support experienced less detrimental impact from the workload on performance. This study verified the negative linkages between mental workload and job performance and identified the boundary conditions (mindfulness and social work support) that weaken the relationship. The limitations and future research directions are then discussed.

**Keywords:** air traffic controller; mental workload; job performance; mindfulness; social work support

## 1. Introduction

ATCOs' job performance not only has an impact on an organization's existence but also directly affects flight safety [1]. As such, job performance among ATCOs has garnered considerable research attention from scholars. Air travel is highly reliable globally and is considered the safest channel of transportation; however, air collisions have happened which have cost innumerable lives and assets. Human errors occur due to various reasons, one of which is increased mental workload [1]. A high level of mental workload decreases the attention of employees and compromises their judgement, as well as their decision making [2]. However, surprisingly, other scholars reported conflicting findings in various contexts. For instance, Pourteimour et al., reported a positive relationship between mental workload and job performance [3]. Additionally, Omolayo and Omole and Susiarty et al.,claimed that mental workload has no impact on job performance [4,5]. This highlights a gap in the literature which necessitate to examining the impact of mental workload on the job performance in the context of air traffic control.

The Civil Air Navigation Services Organization (CANSO) indicated variations in the performance of air traffic services (ATS) in their annual performance reports, which are due to various factors such as on-job training, air traffic management systems, availability of air traffic control units, operational procedures, and management styles. Despite the numerous

types of air transport tools that help ATCOs' mental information processing [6,7], the role of an ATCO is still ranked as an occupation with the highest mental workload within the aviation industry [8]. Mental workload is inevitable in ATCOs' jobs; however, its impact on job performance needs to be managed such that it does not compromise flight safety, knowing that various air navigation services providers have variations in their operational processes and capabilities. The job demand–resources (JDR) model states that professional and personal resources weaken the negative impact of job demands on job performance [9]. Yang, Rantanen, and Zhang indicated that ATCOs with high personal resources (time efficacy) experienced lower mental workload than ATCOs with lower time efficacy [10]. Recent studies highlighted mindfulness and social work support as resources that provide positive outcomes in the workplace [11–13]. Mindfulness refers to the individual's ability to be aware of the current situation and stay concentrated on the job [14]. Social work support is the level of advice and assistance that individuals receive from others while performing a job [15]. However, there are conflicting findings associated with the effect of mindfulness and social work support in the workplace. Multiple researchers suggested that mindfulness might exhibit a negative healthy impact in the workplace [16–19]. Additionally, scholars claimed that social work support negatively impacts the health and well-being of individuals [20–25]. Furthermore, Alaydi et al. recommended the exploration of the role of personal resources such as mindfulness and social work support in the JDR model to elucidate their effects, particularly in aviation [11]. This indicated the need to explore the moderating role of both variables between mental workload and job performance. Thus, this study intends to answer three research questions as follows:

1. Does mental workload negatively impact the ATCOs' job performance?
2. Does mindfulness weaken the mental workload–job performance relationship?
3. Does social work support weaken the mental workload–job performance relationship?

## 2. Literature Review

### 2.1. JDR Theory

Numerous scholars have suggested that the JDR theory is one of the most appropriate theories to explain employees' job performance in relation to job demands and resources [26]. Job demands are related to job aspects that require consistent mental and emotional skills or efforts, which incur psychological or physical costs, while job resources are associated with job aspects which are useful in achieving work goals, reducing job demands and their related costs, and encouraging personal learning and growth [9]. Together, the interplay of job demands and resources may occur in various occupational settings to explain job outcomes such as performance.

The JDR theory has some fundamental assumptions. First, job characteristics can be classified as either job demands or resources. Second, there are dual processes from job characteristics to outcomes; job demands cause health impairment, while job resources develop motivation. Third, job resources may reduce the impact of job demands [27], where personal resources can be included in the JDR model as a moderator between job characteristics and their outcomes. This extension of JDR presents huge flexibility to the model in its usage as a theoretical framework to understand the role of resources. Moreover, Schaufeli and Taris, suggested that an individual uses their own personal attributes, known as personal resources, to manage job demands, and there is a need to investigate their role in the JDR model [26].

### 2.2. Job Performance

Job performance is the behavior of accomplishing a task that supports the organization's goals [28]. Therefore, job performance is the conduct but not the result [29]. Job performance is related to the operation of tasks and performing them well; it is made of various complex activities [30]. Borman and Motowidle categorized job performance into task and contextual categories. Task performance is defined as "the effectiveness with which job incumbents perform activities that contribute to the organization's technical

core either directly by implementing a part of its technological process, or indirectly by providing it with needed materials or services" [28,31]. In the ATC context, monitoring and managing flights through the radar screen to maintain a safe airspace is an example of task performance [32]. Contextual performance encompasses activities that "contribute to organizational effectiveness in ways that shape the organizational, social, and psychological context that serves as the catalyst for task activities and processes" [28]. In the ATC context, providing support to colleagues during the peak time is an example of contextual performance. Both dimensions are applicable to the ATC context [32]. Job performance is influenced by various job demands and job resources. Mental workload is a type of job demand that affects job performance negatively [27]. The following section reviews the mental workload concept.

### 2.3. Mental Workload

Mental workload refers to the individual's mental effort in performing a task [33]. It is related to the information processing theory, which requires skills pertaining to mental domains such as information processing, communicative aspects, spatial perception, decision making, logical reasoning, and human relations [34]. In the ATC context, mental workload is determined by the number of flights under a controller's responsibility and the conflicts among these flights [33], abnormal traffic, high air traffic volume, and unexpected events [35]. With an increased number of flights, the number of conflicts also increases and mental information processing becomes more intense, which requires a high level of focus on the tasks and results in mental workload. Mental workload is considered a multidimensional construct that results from various demands. Valdehita et al. introduced a model of mental workload with four dimensions. These dimensions include cognitive demand, encompassing necessary mental and perceptual activities like thinking, decision making, and remembering, as well as the ease or complexity of the activity; temporal demand, which pertains to the time pressure associated with tasks in terms of receiving, processing, and completion; performance demand, which indicates the level of success in task completion; and emotional demand, which is linked to the health consequences experienced after completing tasks [36].

Although mental worload is reported to have a negative relationship with job performance [2], there are other studies where there is a positive relationship or no relationship. For instance, Pourteimour et al., reported a positive relationship between mental workload and job performance among healthcare employees [3]. Furthermore, Omolayo and Omole indicated the absence of the mental workload–job performance relationship among the academic and non-academic workforce [4]. Susiarty et al. reported no influence of mental workload on job performance among nurses [5].

The ATC job has the highest mental workload in the aviation industry compared to other jobs such as pilots and flight engineers [8]. In fact, Majumdar and Ochieng warned that ATC mental workload is an important indicator that needs to be monitored before it affects the safety of planes [37]. This happens due to the positive relationship between high mental workload and the possibility of human mistakes which create threatening outcomes for aviation safety [38].

### 2.4. Mindfulness

Mindfulness is the practice of being aware of a situation and remaining focused on work [14]. Mindfulness is a personal resource that generates positive outcomes in the workplace; it is an enhancement tool for human well-being [39]. Cardaciotto et al. suggested that mindfulness is composed of two dimensions: acceptance and awareness. Awareness involves continual monitoring experience, focusing on the present rather than being preoccupied with past or future events. Acceptance entails fully embracing a situation with a disposition to accept it as it is. It is suggested that acceptance represents how present-moment awareness is practiced, devoid of judgement, adopting a new attitude of acceptance, openness, and even compassion toward one's experience [40]. Jha indicated that

mindfulness impacted job satisfaction and affective commitment positively via employee behavior [41]. In addition, Ngo et al. reported that mindfulness has a positive direct impact on job performance and an indirect impact via the creative process's engagement and employee creativity [42]. In the ATC context, mindfulness was shown to lead to positive health outcomes, such as improved memory and concentration as well as less irritability, tension, and exhaustion, based on a pilot study conducted among air traffic controllers in Spain [43]. In similar settings, Li et al. reported mindfulness as having a negative influence on pilots' anxiety [44]. Along the same lines, mindfulness was found to increase job satisfaction and decrease emotional exhaustion [45]. Also, Meland et al. reported that mindfulness has a negative influence on somatic anxiety related to performance [46].

However, the effects of mindfulness at work are not always positive. Britton raised the flag regarding the possibility that mindfulness may have an inverted U-shaped curve relationship with performance [16]. For example, mental health issues (substance abuse, anxiety, and increased depression) increase with the increase in mindful attention (observing awareness) that is the core aspect of mindfulness [18]. Therefore, excessive mindfulness is related to the depletion of personal resources such as attention, which is important for individual performance [17]. In addition, Schindler et al. indicated that mindfulness reduces moral responses to an unethical or illegal behavior that causes harm [19]. It is notable that the impact of mindfulness at work varies across different job contexts. Hence, this emphasizes the need to investigate its impact and various roles in the ATC context to leverage its full advantages [11].

*2.5. Social Work Support*

Social work support refers to the extent to which a person receives advice and assistance from others [15]. Behaviors by colleagues and supervisors that are performed to promote employee development are considered as social work support [47]. Social support is a psychosocial resource, where good interactions lead to good social support when needed. However, such support cannot be fully received if an individual's social interaction skills are limited or weak [48]. The positive and healthy influence of social work support in the organization is justified from three perspectives. The first is the behavioral aspect, which suggests that social work support improves health or minimizes the impact of negative factors on health through the encouragement of behavioral changes. Individuals with healthy social work support connections are encouraged to engage in healthy behavior [49] or gain beneficial information about healthy behaviors [50]. The second perspective is the psychological aspect, which argues that individuals perceive that the help provided by others results in positive impacts, thus improving their psychological state and mental and physical health [51]. The third perspective is related to the physiological aspect, which states that social work support relaxes the individual and strengthens the immune system response according to physiological mechanisms [52]. In other words, social work support triggers positive brain system responses to reverse stress [53].

Many research studies have presented a well-established relationship between social work support and psychological and physical health outputs. A general benefit of social work support is its ability to provide positive experiences and rewarding acts in society. Social support may be related to comfort due to its association with positive feelings and a recognition of self-worth. Such a sort of support can also lead to general well-being for the reason that it engenders a positive effect, feelings of predictability and stability in an individual's life condition, and an identification of self-worth. Furthermore, such support offers social network integration, which assists in avoiding unpleasant experiences such as legal or economic problems [54].

Social support plays an important role in the human resource context due to its impact in an organization. Mixed findings have been reported on the relationship between social work support and employee behavior. Most researchers have found a positive relationship between them. For example, it is indicated that social work support from co-workers improves job performance among employees [55]. Social support also has a direct positive

relationship with work engagement [56]. Specifically, the social support (organizational, supervisory, and colleagues)—job performance relationship is mediated by work engagement, though only colleagues' social support directly enhances job performance [56]. Correspondingly, Hoc et al. indicated that colleagues' social support weakens the relationships between turnover intention, perceived customer relations, and job stress, especially for the relationship between job stress and turnover intention [57]. In addition, social support and control hinder strong negative feelings toward one's job [58].

However, this positive relationship contradicted other studies' results. The results of other studies are as follows: (1) social work support has an unhealthy impact on individuals' well-being and health [20,23]; (2) individuals with low social work support and high workload have less stress than individuals with high social work support [25]; (3) individuals with the most complex tasks and less social work support are more satisfied with their jobs than those with high social work support [21]; (4) employees with higher organizational support are more likely to report adverse effects on their health from their place of work [24]; and (5) social work support leads to counterproductive behaviors [22].

Scholars have attempted to explain the negative impact of social work support in the workplace. One explanation is related to the stressful nature of social relations that require energy and time to develop and maintain [59]. Also, it was suggested that depression may occur if social work support becomes ineffective because it stimulates feelings of helplessness in the supposed receiver of the support [60]. Another explanation is that support may be a distraction from concentration during a task, leading to incomplete or overdue tasks [23]. In addition, it is suggested that the negative impact of social work support results from the attitude that receiving support is a sign of weakness [61]. Unhealthy phenomena such as sickness, absence, or complaints may also be triggered by high social support and interactions because they indicate that such behaviors are appropriate and acceptable in the organization [62].

The role of social support is thus diverse in job stress management [63]. It can be a mediator, an explanatory variable, or a moderator, although its buffering impact is uncertain. Also, it can act as an antecedent of stressors or an outcome of stress [64]. This shows that the role of social work support at work is controversial. Therefore, there is an important need to revisit the impact of social support as a moderating variable [57].

## 3. Hypothesis Development and Research Framework

### 3.1. Hypothesis Development

There has been a growing interest in understanding the consequences of mental workload on organizational results. ATC is viewed as a demanding profession characterized by a challenging and extreme workload [65]. Empirical studies in the ATC context found that mental workload has a direct negative impact [66,67], and an indirect impact through stress [35], on job performance. Chuang et al. also found that ATCs who have experienced very stressful long-term working conditions demonstrate negative physical and psychological effects that easily lead to poor performance [68]. The JDR model classifies mental workload as one of the aspects related to job demands, because when mental workload increases, additional efforts are required to complete a job. In such scenarios, high job demands can negatively impact employees' work performance [27]. Therefore, the present research hypothesizes that when air traffic controllers are confronted with increased mental workload, their job performance suffers:

**H$_1$:** *Mental workload has a negative relationship with job performance.*

Mindfulness can play a moderating role in the relationship between mental workload and job performance. There is a clear affirmation that mindfulness impacts job performance positively and buffers the health impairment process in the JDR. Mindfulness is reported to facilitate job performance [42,69] Furthermore, Guidetti et al. indicated that mindfulness reduces the negative impact of the workload on burnout [13]. With an increase in mindfulness level, the positive workload stress appraisal–burnout relationship decreases. In the

ATC context, those who are mindful and aware of current scenarios in flight movements may be less bothered by mental workload. Therefore, mindfulness is likely to reduce the negative impact of mental workload on performance, as hypothesized below:

**H2:** *Mindfulness moderates the mental workload–job performance relationship, such that at a high level of mindfulness, the negative relationship is weaker.*

Notably, several studies have employed social work support as a moderator and reported desirable results. First, social work support weakens the relationship between job complexity and job stress [70], as knowledge shared from a colleague may help employees manage stressful situations. Second, it moderates the relationship between role stress and turnover intentions by reducing the turnover intention of workers suffering from role stress [71]. Third, Zhou and Lin found that individuals with high levels of social support demonstrate a stronger positive adaptability–life satisfaction relationship among freshmen Chinese students at a university in China [72]. Fourth, it moderates the negative relationship between workload and performance among academic professionals [48], as well as the psychological capital–engagement relationship of students [12].

In the case of ATC, social work support given by co-workers could allay the impact of mental workload on performance. For instance, in the case of unresolved work challenges for an air traffic controller, the willingness of co-workers to provide support and help resolve the issue together could enhance the air traffic controller's confidence in solving similar challenges in the future, which then leads to improved performance. Considering that most empirical studies have reported the usefulness of social work support as a job resource within the JDR, this study hypothesis that ATCOs who experience social work support would demonstrate less workload impact on performance. Thus, H3 is developed as below:

**H3:** *Social work support moderates the mental workload–job performance relationship, such that at a high level of social work support, the negative relationship is weaker.*

*3.2. The Research Framework*

Underpinned by the JDR model, this paper proposes that ATCOs' mental workload, being the key to job demand, induces a negative impact on job performance. This negative relationship may be buffered by personal resources (mindfulness) and job resources (social work support). Figure 1 shows the research.

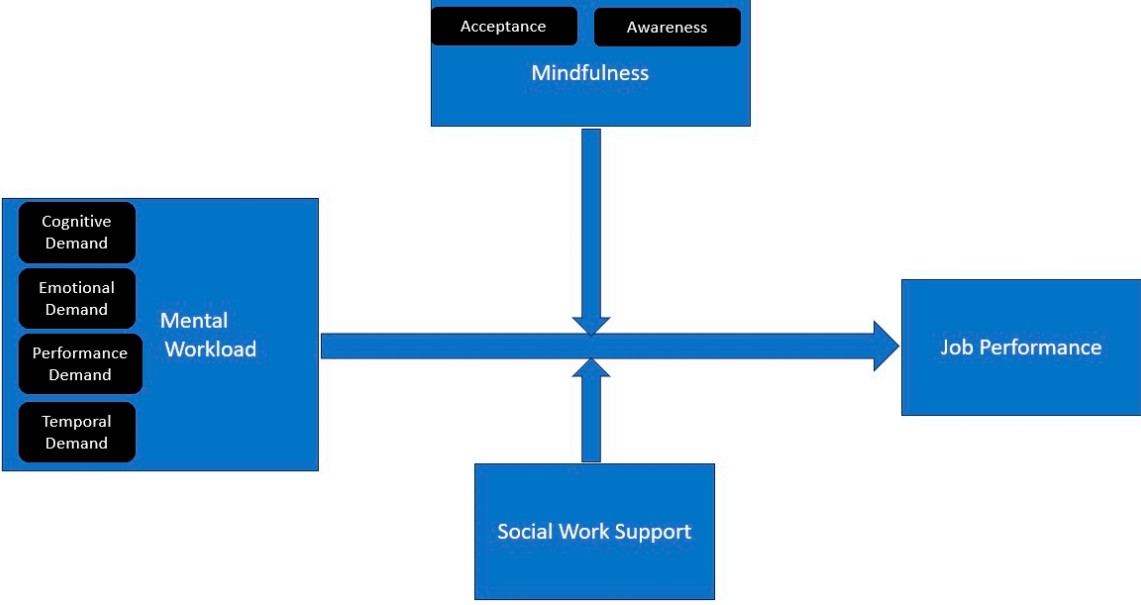

**Figure 1.** Research framework.

## 4. Research Methodology

### 4.1. Respondents and Measurements

A cross-sectional survey approach was utilized, using face to face data collection method. ATCOs working for the Saudi Air Navigation Services company (SANS), which manages 13 ATC units, were invited to participate in the questionnaire survey. A total of 450 questionnaires were distributed, and 324 completed questionnaires were returned.

### 4.2. Measures

All measures were adapted from previous studies. Mental workload was measured using four dimensions: cognitive (10 items), emotional (7 items), temporal (7 items), and performance (5 items). The reliability of the scale is 0.8 [36]. Mindfulness scale is a 10-item scale consisting of two dimensions: awareness (five items, Cronbach's $\alpha$ = 0.90) and acceptance (five items, Cronbach's $\alpha$ = 0.93) [73]. Social work support scale has 6 items, with a Cronbach's $\alpha$ = 0.82 [15]. Job performance scale has two dimensions: task performance (7 items, Cronbach's $\alpha$ = 0.82) and contextual performance (12 items, Cronbach's $\alpha$ = 0.90) [74]. It should be noted that many researchers supported the use of job performance self-assessment and reported that it is reliable and has been widely used in numerous studies [75–77].

## 5. Results

### 5.1. Findings

The scale of a negative item (AC1-5) was reverse-coded to ensure all measurement items followed the same direction [78]. Next, the results of the analysis with the Harman's single factor technique indicated that the explained variance of the first factor was 18.06% (below the threshold limit of 50%). This indicated that common method variance is not a significant issue [79]. As can be seen in Table 1, the majority of the respondents were male (94.8%), aged between 26 to 35 (61.2%), married (70.4%), worked for 10 years or below (66.1%), and worked in the Jeddah or Riyadh unit (62.99%).

**Table 1.** Demographic profile.

| Description | | Frequency | Percent |
|---|---|---|---|
| Gender | Male | 307 | 94.8 |
| | Female | 17 | 5.20 |
| Age | 25 years old and below | 28 | 8.60 |
| | 26–30 years old | 113 | 34.90 |
| | 31–35 years old | 82 | 25.30 |
| | 36–40 years old | 34 | 10.50 |
| | 41–45 years old | 35 | 10.80 |
| | 46 years old and above | 32 | 9.90 |
| Marital Status | Single | 88 | 27.20 |
| | Married | 228 | 70.40 |
| | Divorce | 8 | 2.50 |
| Education | Diploma | 155 | 47.80 |
| | Bachelor's Degree | 149 | 46.00 |
| | Master's Degree or higher | 20 | 6.20 |
| Working Experience | Less than 5 years | 99 | 30.60 |
| | 5–10 years | 115 | 35.50 |
| | 11–15 years | 29 | 9.00 |
| | More than 15 years | 81 | 25.00 |
| Job Position | Tower | 132 | 40.70 |
| | Approach | 97 | 29.90 |
| | Area | 95 | 29.30 |

| Description | | Frequency | Percent |
|---|---|---|---|
| Unit | Jeddah | 124 | 38.27 |
| | Riyadh | 80 | 24.70 |
| | Dammam | 32 | 9.90 |
| | Madinah | 10 | 3.09 |
| | Abha | 16 | 4.90 |
| | Hail | 9 | 2.80 |
| | Alhasa | 4 | 1.20 |
| | Jazan | 13 | 4.00 |
| | Qasim | 12 | 3.70 |
| | Tabouk | 7 | 2.20 |
| | Taif | 11 | 3.40 |
| | Yanbu | 6 | 1.90 |
| | Najran | 0 | 0 |
| | Total | 324 | 100.00 |

*5.2. Evaluation of Measurement Model*

To assess the measurement model, it is necessary to examine the psychometric characteristics of the constructs: construct reliability, convergent validity, and discriminant validity [80]. Composite reliability was employed to assess the reliability. Using Joreskog's guideline, all the constructs demonstrated composite reliability scores ranged between 0.838 and 0.987 (see Table 2), which is considered adequate as they surpassed the minimum threshold value of 0.70 [81]. Factor loading and average variance extracted were used to evaluate the adequacy of the convergent validity. The results indicate that the loading of all items meets the suggested value of 0.40 or more [82]. While, for the average variance extracted (AVE), all constructs illustrated adequate values between 0.512 and 0.917 (above the threshold limit of 0.50), for temporal demand, two items of temporal demand (i.e., TD1 = 0.535 and TD7 = 0.469) with low loading values were deleted to achieve the minimum score of 0.50 for AVE [83].

**Table 2.** Results of measurement model.

| | Mean | SD | O/L | CR | AVE |
|---|---|---|---|---|---|
| Acceptance | | | | 0.89 | 0.618 |
| AC1; I try to distract myself when I feel unpleasant emotions. | 3.938 | 0.96 | 0.777 | | |
| AC2; I try to stay busy to keep thoughts or feelings from coming to mind. | 3.966 | 1.022 | 0.807 | | |
| AC3; I tell myself that I shouldn't feel sad. | 3.954 | 1.01 | 0.822 | | |
| AC4; If there is something I don't want to think about, I'll try many things to get it out of my mind. | 4.108 | 0.989 | 0.827 | | |
| AC5; When I have a bad memory, I try to distract myself to make it go away. | 3.991 | 0.935 | 0.69 | | |
| Awareness | | | | 0.838 | 0.512 |
| AW1; I am aware of what thoughts are passing through my mind | 3.787 | 0.681 | 0.618 | | |
| AW2; When someone asks how I am feeling, I can identify my emotions easily. | 3.728 | 0.672 | 0.592 | | |
| AW3; I am aware of thoughts I'm having when my mood changes. | 3.793 | 0.701 | 0.805 | | |
| AW4; Whenever my emotions change, I am conscious of them immediately. | 3.756 | 0.801 | 0.815 | | |
| AW5; When talking with other people, I am aware of the emotions I am experiencing. | 3.769 | 0.885 | 0.719 | | |
| Cognitive demand | | | | 0.94 | 0.613 |
| CD1; My work involves the processing of complex information | 2.102 | 0.706 | 0.692 | | |
| CD2; My job requires thinking and choosing between different alternatives | 2.278 | 0.655 | 0.783 | | |
| CD3; I have to make difficult decisions. | 2.225 | 0.64 | 0.706 | | |
| CD4; My job requires handling a lot of knowledge. | 2.052 | 0.699 | 0.869 | | |
| CD5; My job requires dealing with information that is perceived with difficulty. | 1.522 | 1.007 | 0.765 | | |
| CD6; I have to deal with information that is not easily understood. | 1.272 | 0.899 | 0.803 | | |
| CD7; My job requires a lot of information. | 1.867 | 0.76 | 0.773 | | |
| CD8; My job requires memorizing a high amount of data. | 1.528 | 0.84 | 0.861 | | |
| CD9; My work is mentally intense. | 2.074 | 0.589 | 0.731 | | |
| CD10; I have to do a great search and information gathering to carry out my tasks. | 1.142 | 0.935 | 0.826 | | |

**Table 2.** *Cont.*

| | Mean | SD | O/L | CR | AVE |
|---|---|---|---|---|---|
| Emotional Demand | | | | 0.987 | 0.917 |
| ED1; I have trouble forgetting the problems of my job | 1.154 | 1.028 | 0.908 | | |
| ED2; My work makes me nervous. | 1.102 | 1.065 | 0.939 | | |
| ED3; My work is affecting my personal relationships (family, friends...). | 0.966 | 1.078 | 0.942 | | |
| ED4; I feel very tired, physically fatigued. | 1.198 | 0.964 | 0.982 | | |
| ED5; My work affects me a lot emotionally. | 1.201 | 0.988 | 0.968 | | |
| ED6; When I finish my workday, I feel a lot of physical exhaustion. | 1.201 | 0.959 | 0.978 | | |
| ED7; My work is affecting my health | 1.225 | 1.001 | 0.983 | | |
| Contextual Performance | | | | 0.94 | 0.568 |
| CP1; I took on extra responsibilities. | 4.056 | 0.833 | 0.727 | | |
| CP2; I started new tasks myself, when my old ones were finished. | 4.025 | 0.835 | 0.735 | | |
| CP3; I took on challenging work tasks, when available. | 4.185 | 0.88 | 0.832 | | |
| CP4; I worked at keeping my job knowledge up-to-date. | 4.136 | 0.892 | 0.725 | | |
| CP5; I worked at keeping my job skills up-to-date. | 4.201 | 0.839 | 0.786 | | |
| CP6; I came up with creative solutions to new problems. | 4.222 | 0.805 | 0.767 | | |
| CP7; I kept looking for new challenges in my job. | 4.299 | 0.835 | 0.823 | | |
| CP8; I did more than was expected of me. | 4.012 | 0.824 | 0.744 | | |
| CP9; I actively participated in work meetings. | 3.972 | 0.873 | 0.709 | | |
| CP10; I actively looked for ways to improve my performance at work. | 4.216 | 0.822 | 0.767 | | |
| CP11; I grasped opportunities when they presented themselves. | 4.262 | 0.833 | 0.806 | | |
| CP12; I knew how to solve difficult situations and setbacks quickly. | 4.099 | 0.814 | 0.597 | | |
| Task Performance | | | | 0.928 | 0.648 |
| TP1; I managed to plan my work so that it was done on time. | 4.052 | 0.861 | 0.805 | | |
| TP2; My planning was optimal. | 4.114 | 0.873 | 0.834 | | |
| TP3; I kept in mind the results that I had to achieve in my work. | 4.049 | 0.866 | 0.798 | | |
| TP4; I was able to separate main issues from side issues at work. | 3.954 | 0.917 | 0.834 | | |
| TP5; I knew how to set the right priorities. | 4.025 | 0.926 | 0.822 | | |
| TP6; I was able to perform my work well with minimal time and effort. | 3.559 | 1.232 | 0.83 | | |
| TP7; Collaboration with others was very productive. | 4.262 | 0.833 | 0.705 | | |
| Performance demand | | | | 0.954 | 0.805 |
| PD1; My job requires maintaining a high level of attention. | 2.565 | 0.603 | 0.871 | | |
| PD2; My job requires no mistakes. | 2.417 | 0.6 | 0.835 | | |
| PD3; I have to give very precise responses | 2.611 | 0.585 | 0.952 | | |
| PD4; My mistakes can have serious consequences. | 2.645 | 0.578 | 0.915 | | |
| PD5; My job involves a lot of responsibility. | 2.682 | 0.562 | 0.908 | | |
| Social work support | | | | 0.886 | 0.568 |
| SWS1; I have the opportunity to develop close friendships in my job. | 3.944 | 0.877 | 0.576 | | |
| SWS2; I have the chance in my job to get to know other people. | 3.599 | 1.127 | 0.837 | | |
| SWS3; I have the opportunity to meet with others in my work. | 3.651 | 1.122 | 0.836 | | |
| SWS4; My supervisor is concerned about the welfare of the people that work for him/her | 3.997 | 0.97 | 0.824 | | |
| SWS5; People I work with take a personal interest in me. | 4.222 | 0.716 | 0.722 | | |
| SWS6; People I work with are friendly. | 4.287 | 0.725 | 0.692 | | |
| Temporal demand | | | | 0.859 | 0.55 |
| TD1; I have to work constantly; I cannot take breaks beyond strict regulations. | | D | | | |
| TD2; The pace of work is excessive, difficult to reach even by an experienced worker. | 1.815 | 0.851 | 0.649 | | |
| TD3; I often work with annoying interruptions | 1.352 | 0.959 | 0.683 | | |
| TD4; I cannot stop my work when I need it. | 2.444 | 0.741 | 0.765 | | |
| TD5; The pace of work is imposed on me. | 2.25 | 0.704 | 0.818 | | |
| TD6; The accomplishment of my tasks demands a lot of speed. | 2.401 | 0.732 | 0.781 | | |
| TD7; It is normal for me to accumulate the tasks. | | D | | | |

D—Item deleted due to low loading (<0.40), SD—standard deviation; CR—composite reliability; AVE—average variance extracted. To affirm the reflective constructs' discriminant validity, heterotrait–monotrait ratio of correlations (HTMT) was applied. As indicated in Table 3, all reflective constructs exhibited acceptable discriminant validity, as HTMT values were lower than the conservative threshold limit of 0.85 [84]. Overall, it can be concluded that all the constructs were truly different from each other.

**Table 3.** Results of HTMT.

| | AC | AW | CD | CP | ED | PD | SW | TD | TP |
|---|---|---|---|---|---|---|---|---|---|
| AC | | | | | | | | | |
| AW | 0.248 | | | | | | | | |
| CD | 0.077 | 0.066 | | | | | | | |

**Table 3.** *Cont.*

|  | **AC** | **AW** | **CD** | **CP** | **ED** | **PD** | **SW** | **TD** | **TP** |
|---|---|---|---|---|---|---|---|---|---|
| CP | 0.255 | 0.261 | 0.192 | | | | | | |
| ED | 0.052 | 0.049 | 0.531 | 0.217 | | | | | |
| PD | 0.077 | 0.08 | 0.142 | 0.199 | 0.212 | | | | |
| SW | 0.464 | 0.414 | 0.145 | 0.444 | 0.086 | 0.062 | | | |
| TD | 0.091 | 0.101 | 0.477 | 0.181 | 0.451 | 0.804 | 0.083 | | |
| TP | 0.269 | 0.146 | 0.092 | 0.337 | 0.093 | 0.172 | 0.264 | 0.152 | |

HTMT < 0.85.

### 5.3. Evaluation of Higher-Order Constructs

The research has constructs conceptualized as reflective–formative higher-order constructs (HOCs) that were made up of several lower-order constructs (LOCs). Mindfulness contains two LOCs: acceptance and awareness [73]. Mental workload has four LOCs: cognitive, performance, temporal, and emotional [36]. Job performance has two LOCs: task and contextual [32]. As suggested by Sarstedt et al., all the HOCs were evaluated using a two-stage approach [85]. Firstly, all the LOCs were evaluated using the regular procedure of the reflective measurement model, as reported earlier. Secondly, the HOCs were evaluated using the regular procedure of the formative measurement model.

Initially, the variance inflation factor (VIF) was utilized to examine the formatively measured constructs collinearity. VIF values for all LOCs were between 1.056 and 2.745 and below 3.0 [80], confirming no collinearity issues (see Table 4). Subsequently, each LOC significance and outer weights was examined through the bootstrapping technique with 5000 re-samples. All LOCs for mindfulness (i.e., acceptance, awareness), job performance (i.e., contextual performance, task performance), and mental workload (i.e., cognitive, temporal, performance, emotional) were statistically significant at $p < 0.01$.

**Table 4.** Results of higher-order constructs.

| **HOC** | **LOC** | **Outer Weight** | **Standard Deviation** | **VIF** | **t-Statistics** | ***p* Values** |
|---|---|---|---|---|---|---|
| MIND | AC | 0.678 | 0.056 | 1.056 | 14.294 | 0 |
| | AW | 0.595 | 0.077 | 1.056 | 5.885 | 0 |
| | CD | 0.686 | 0.024 | 1.852 | 19.376 | 0 |
| MW | TD | −0.566 | 0.017 | 2.745 | 9.98 | 0 |
| | ED | 0.305 | 0.022 | 1.486 | 23.464 | 0 |
| | PD | 0.994 | 0.032 | 2.034 | 4.73 | 0 |
| JP | TP | 0.274 | 0.040 | 1.106 | 9.04 | 0 |
| | CP | 0.88 | 0.037 | 1.106 | 22.262 | 0 |

### 5.4. Evaluation of Structural Model and Moderation Analysis

The bootstrapping technique results (Table 5) demonstrated that mental workload was negatively related to job performance ($H_1$: β = −0.264, t = 5.609, $p < 0.001$) significantly; thus, $H_1$ was supported. For moderation analysis, interaction terms were developed based on a two-stage approach [86]. Using a bootstrapping re-sampling technique of 5000 re-samples [87], there is a significant moderating effect of mindfulness on the path between MW and job performance (β = 0.130, t = 2.302, $p = 0.011$). Using the interaction plot introduced by Dawson, it can be seen that at a high level of mindfulness, the negative relationship between MW and job performance becomes weaker, providing support to $H_2$ (see Figure 2) [88]. Also, there is a significant moderating effect of social work support on the path between MW and job performance (β = 0.161, t = 2.147, $p = 0.016$). The interaction plot shows that at a high level of social work support, the negative relationship between MW and job performance becomes weaker, providing support to $H_3$ (see Figure 3).

**Table 5.** Results of structural model.

| | Std Beta | Standard Error | t-Statistics | *p* Values |
|---|---|---|---|---|
| Direct Relationship | | | | |
| MW -> JP | −0.264 | 0.047 | 5.609 | 0 |
| Moderation Relationships | | | | |
| MINDF-MW -> JP | 0.130 | 0.056 | 2.302 | 0.011 |
| SWS-MW -> JP | 0.161 | 0.075 | 2.147 | 0.016 |

MW: mental workload; MINDF: mindfulness; SWS: social work support; JP: job performance.

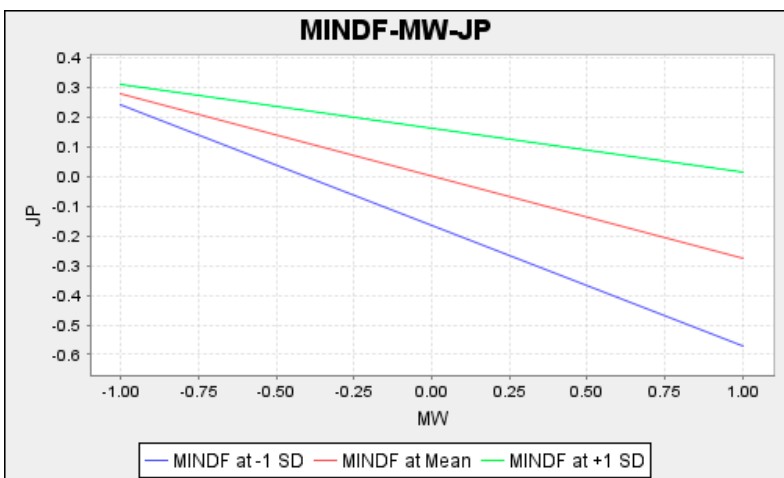

**Figure 2.** The interaction plot of mindfulness on the path between MW and job performance.

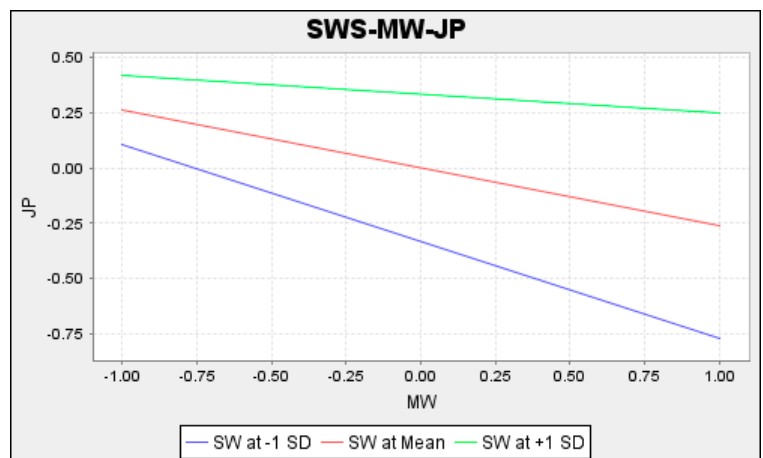

**Figure 3.** The interaction plot of social work support on the path between MW and job performance.

## 6. Discussion

The finding of the analysis shows that mental workload has a significant relationship with job performance (H₁ was supported; mental workload has a negative relationship with job performance). This finding is in line with the job demand–resources model, that job demands bring a negative work outcome. The findings also support most of the previous studies. For instance, Majumdar and Ochieng, Pereira, and Edwards et al. indicated that mental workload reduces job performance directly [37,66,67]. Similarly, Pant et al. evidenced this negative relationship and commented that the work associated with a high mental workload generally requires a high level of focus in ensuring flights' safely. Human errors are expected to occur if the mental workload exceeds individual capability [33].

The findings show a high mindfulness level weakens the negative impact of mental workload on job performance (H₂ was supported; mindfulness moderates the mental workload–job performance relationship, such that at a high level of mindfulness,

the negative relationship is weaker). This finding is in line with the notion of the job demand–resources model that personal resources develop positive outcomes in the health impairment process by weakening the negative influence of job demands on job performance [9]. Also, it supports the majority of empirical studies that have reported the healthy effect of mindfulness in an organization [69,89]. Within the ATC context, mindfulness is air traffic controllers' full concentration on flights' movement and living a situation fully, with the attitude of accepting it as it is, combining the two dimensions together. It is the ability of ATCOs to be calm and unstressed or confused during abnormal flight movements, e.g., weather conditions, increased flights, or equipment failure, which could cause increases in the mental workload. Air traffic controllers who have their full concentration on flight movements and stay calm during an increase in flights perform better.

As predicted, social work support buffers the negative impact of mental workload on job performance (H3 was supported; social work support moderates the relationship between mental workload and job performance, such that at a high level of social work support, the negative relationship is weaker). This means that the job performance of individuals with high social work support is less affected by mental workload, as supportive co-workers alleviate the mental workload's impact. In contrast, the mental workload's negative effect on performance is stronger among those with low social work support. This finding is in line with the JDR theory's statement that work resources (i.e., social work support) buffer the health impairment processes [90]. Furthermore, it supports the findings of previous studies. For instance, Naqvi et al. introduced social support as a moderating variable between job complexity and job stress, revealing that social support weakens this relationship [70]. Similarly, Kim and Stoner found that social support moderates the relationship between job stress and turnover intentions [71]. Social work support has been found to moderate the negative relationship between workload and performance in the context of education [48]. In the case of ATC, social work support from co-workers weakens the effect of mental workload on job performance. For instance, if a work challenge cannot be resolved by an air traffic controller, co-workers' willingness to help resolve the issue together enhances the air traffic controller's confidence in solving similar challenges in future, which then leads to improved performance.

## 7. Conclusions, Implications, Limitations, and Suggestions for Future Studies

This study adds value to the ATC literature and the job demand–resources model in two ways. First, it provides empirical evidence as to the negative relationship between mental workload and job performance in an ATC context, despite mixed findings across different jobs. Second, this study extends the job demand–resources model by introducing mindfulness as a moderator in the aviation context and finds that mindfulness is applicable in an ATC context, despite controversial curvilinear arguments in the literature. Third, this study adds value to the job demand–resources model by introducing social work support as a conditional factor that reduces workload's impact on job performance.

This study offers insightful ideas to the ATC sector, mainly on ways to enhance ATC performance for safe flight operation. First, the significant negative effect between mental workload and job performance suggests the need to ease the mental workload of ATCOs. ATC management are recommended to conduct intensive and targeted job training for air traffic controllers, which helps to enhance work skills, to increase cognitive abilities, and improve decision making so as to work more efficiently and reduce mental workload [91]. ATC management are advised to restructure the aerospace optimally by changing existing sectors, and combining or de-combining sectors [92], aiming to set up smooth sectors with optimal capacity. Also, management is advised to explore job aspects that can be handled by artificial intelligence and/or introduce air traffic flow management [93].

Since mindfulness and social work support are helpful in reducing mental workload's impact on job performance, ATC human resource managers are suggested to consider introducing mindfulness training sessions in the workplace to gain its positive outcomes [43]. By doing so, mental ability would improve and the impact of mental workload on performance

would be reduced. Furthermore, to promote social work support, the ATC management is recommended to create a friendly and supportive culture where all members are able to work together harmoniously.

Although this study provides considerable insights to both theoretical and managerial perspectives on ATC, it has some limitations that should be acknowledged. First, this study only focused on three factors that affect job performance. Future studies are highly advised to consider other potential variables such as personality traits, emotional intelligence, and benevolent leadership style. Second, this study used a self-assessment technique to measure job performance constructs. Although various studies support that self-ratings or self-assessments are widely accepted as a reliable tool to measure job performance quantitatively [75,76], it is undeniably less biased if supervisors assess their subordinates' performance. Future studies are thus advised to measure job performance from the supervisor's perspective. Third, this study employed a cross-sectional approach. Future research efforts are highly advised to employ a longitudinal study, especially to verify the long-term effect of mindfulness, since some have argued that it may demonstrate a curvilinear effect on performance.

**Author Contributions:** Conceptualization, B.A. and S.-I.N.; methodology, B.A.; software, B.A.; validation, B.A. and S.-I.N.; formal analysis, B.A.; investigation, S.-I.N.; resources, S.-I.N.; data curation, B.A.; writing—original draft preparation, B.A.; writing—review and editing, S.-I.N.; visualization, B.A.; supervision, S.-I.N.; project administration, S.-I.N.; funding acquisition, None. All authors have read and agreed to the published version of the manuscript.

**Funding:** This research received no external funding.

**Institutional Review Board Statement:** The study was conducted in accordance with the Declaration of Helsinki and approved by the Institutional Review Board (or Ethics Committee) of Research Involving Human Subjects of University Putra Malaysia (protocol code JKEUPM-2021–043 and date of approval 22 July 2021).

**Informed Consent Statement:** Informed consent was obtained from all subjects involved in the study.

**Data Availability Statement:** The data presented in this study are available on request from the corresponding author.

**Acknowledgments:** The authors thank PSU for their support.

**Conflicts of Interest:** The authors declare no conflicts of interest.

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
