# Peer review of "Mitigating the Negative Effect of Air Traffic Controller Mental Workload on Job Performance: The Role of Mindfulness and Social Work Support"

_safety, 2023_

Round 1

Reviewer 1 Report

Comments and Suggestions for Authors

The paper is nicely written but there are some things that can be improved:
1. Please explain the underlying theory that supports the needs to investigate mindfulness and social work support. Why both variables are investigated?

2. Please provide a clear representation of the main constructs (mindfulness, SWS, performance, workload) and the respective indicators of each construct.

Comments on the Quality of English Language

1. Please check the F2 in Table 2.
2. Please check the font
3. Please check the grammatical error throughout the manuscript

Reviewer 2 Report

Comments and Suggestions for Authors

1. This study investigated 324 air traffic controllers using various subjective scales to evaluate the impact of mental load, mindfulness, and social support on their job performance. Based on the research objectives of this study, this study only includes subjective performance scale tests, which cannot effectively reflect the actual work performance of air traffic controllers. Therefore, it is recommended to increase the actual work performance of air traffic controllers in order to obtain more reliable conclusions.

 2. In the introduction section of the article, the author only briefly explained that air traffic controllers are in a high workload state, and suggested that the author write in "3 Research Methodology "adds an overview of the tasks of air traffic controllers and briefly describes their operational processes.

 3. The structure of this paper is somewhat unclear: firstly, sections 3.1 to 3.3 belong to the research method, while sections 3.4 to 3.6 belong to the research results, but the author classifies them as research methods; Secondly, there is no "summary" section at the end of the paper; Finally, it is recommended that the author reorganize the structure of the paper and refer to other literature structures in other journals.

4. There are also some detailed errors in this paper:

The font size of lines 142, 234, and 235 does not match;

The font size of Table 2 on page 9 is inconsistent with that of the continuation table on pages 10 and 11, and the format of the table is not aligned;

The "Discussion" in line 361 is on the same line as the caption in Figure 3.

The format of references should be consistent with the requirements of the journal

Suggest the author to revise the paper format again

Comments on the Quality of English Language

 Suggest the author to revise the paper format again

Reviewer 3 Report

Comments and Suggestions for Authors

The article is well-written and focused on an interesting topic. 

There are only minor issues like:

- research methodology covers results, which is confusing; these shall be divided into two chapters;

- face-to-face data collection method or distributing questionnaires; both cannot be true;

- the formatting of the "Discussion" heading is wrong

- all references shall be thoroughly checked as some did not match (e.g. (Gabriela & Carmen, 2016) is not in the reference list, (Hermosilla et al., 2020) is not in the reference list...)

Reviewer 4 Report

Comments and Suggestions for Authors

Abstract

1. The research objectives have not been clearly defined. In the case of this study, there should be a logical sequence on the stated objectives. The feasibility to carry out one of the objective in the study would depend on the outcome of another objective. "...this study intends to identify conditional factors that could mitigate the negative mental workload impact on ATCOs’ job performance." This objective can only be carried out if there is a negative correlation between the two fixed variables (mental workload and job performance). The second objective could look at the impact of variable such as mindfulness and social work support on the correlation. Describe the objectives in a logical sequence.

2. What has been found in the literature on similar studies? What gaps did you find in the literatures that required this study to be carried out to close those gaps? This domain has been studied for decades. Hence, this point is crucial.

Introduction

1. Please cite the original ICAO source document for this - "...the International Civil Aviation Organization (ICAO) argued that increased workload in the ATC which corresponds to mental work-load has no impact on ATCOs’ performance". (line 40)

2. "the performance of Air Traffic Services (ATC) in their annual performance reports, which..."(line 48) - correct the abbreviation, it should be ATS.

3. "Despite 50 the numerous types of air transport technology that helps ATCOs’ mental information processing..."(line 51) - the term 'air transport technology' here does not reflect the purpose in this sentence, consider changing it to 'ATC tools' or a more suitable one.

4. I have noticed the Authors have been using the term 'mental workload' and 'job demands' interchangeably referring to the same. They are not the same. Job demand can refer to other types of workload. Consider being consistent in the usage, or define in the first attempt that the study will refer the term only for mental workload.

5. "They are promising resources that reduce the unhealthy effect  of mental workload on job performance."(line 67) - How did Authors came to this conclusion at this point ? Is this from the literature? If yes cite. Otherwise, this can only be stated after the analysis and findings section.

6. "Does mindfulness weaken workload-performance relationship? Does social work support weaken workload-performance relationship?" (line 71 & 72) - Be precise, is it 'workload' or 'mental workload'?

Literature

1. Why is the research framework placed under the Literature section? 

2. Figure 1 does not show any meaningful framework. Think of a more meaningful way to present the framework.

3. In section 2.6, other researchers have already answered the research questions drafted in the section. Are the Authors trying to replicate the studies to validate the results with bigger samples? Explain this before the coming up with the framework. Consider placing the Research Framework in a new section.

Conclusion

Develop a Conclusion section to verify the hypotheses developed. Line 362-405 can be placed under the Conclusion section.

Suggestion

The Authors could include more variables based on the data collected to determine the impact on the mental workload - job performance. This will enhance the study from existing studies. 

Comments on the Quality of English Language

No major issues identified with the language. Minor edits are required.

Round 2

Reviewer 1 Report

Comments and Suggestions for Authors

Where are the acceptance, awareness, cognitive demand, emotional demand, and temporal demand constructs in the framework?

Author Response

the dimensions of the constructs are added to the research framework.

thank you for reviewing the research and providing valuable comments in the 1st and 2nd round which improved the research.

Reviewer 2 Report

Comments and Suggestions for Authors

There is no suggestion further. 

Comments on the Quality of English Language

There is no suggestion further. 

Author Response

Thank you for reviewing my research

Reviewer 3 Report

Comments and Suggestions for Authors

The paper is fine. There is a missing end on line 151.

Author Response

the missing end in line 151 has been fixed.

thank you for your valuable comments in round 1 and 2

Reviewer 4 Report

Comments and Suggestions for Authors

Thank you for improving the manuscript based on the comments provided.

Author Response

Thank you for reviewing my research

you provided valuable comments which improved the research